# The Ameliorative Effect of Dexamethasone on the Development of Autoimmune Lung Injury and Mediastinal Fat-Associated Lymphoid Clusters in an Autoimmune Disease Mouse Model

**DOI:** 10.3390/ijms23084449

**Published:** 2022-04-18

**Authors:** Yaser Hosny Ali Elewa, Md Abdul Masum, Sherif Kh. A. Mohamed, Md Rashedul Islam, Teppei Nakamura, Osamu Ichii, Yasuhiro Kon

**Affiliations:** 1Laboratory of Anatomy, Department of Basic Veterinary Sciences, Faculty of Veterinary Medicine, Hokkaido University, Hokkaido 060-0818, Japan; masum@vetmed.hokudai.ac.jp (M.A.M.); rashed.suth@sau.edu.bd (M.R.I.); nakamurate@vetmed.hokudai.ac.jp (T.N.); ichi-o@vetmed.hokudai.ac.jp (O.I.); y-kon@vetmed.hokudai.ac.jp (Y.K.); 2Department of Histology and Cytology, Faculty of Veterinary Medicine, Zagazig University, Zagazig 44511, Egypt; 3Department of Anatomy and Embryology, Faculty of Veterinary Medicine, Zagazig University, Zagazig 44511, Egypt; sherifanatomy81@gmail.com; 4Department of Biological Safety Research, Chitose Laboratory, Japan Food Research Laboratories, Hokkaido 066-0052, Japan; 5Laboratory of Agrobiomedical Science, Faculty of Agriculture, Hokkaido University, Hokkaido 060-0818, Japan

**Keywords:** autoimmune disease mouse model, dexamethasone, high endothelial venules, lung injury, lymphatic vessels, mediastinal fat-associated lymphoid cluster, proliferating cells

## Abstract

In our previous study, we revealed the ameliorative therapeutic effect of dexamethasone (Dex) for Lupus nephritis lesions in the MRL/MpJ-*Fas ^lpr/lpr^* (Lpr) mouse model. The female Lpr mice developed a greater number of mediastinal fat-associated lymphoid clusters (MFALCs) and inflammatory lung lesions compared to the male mice. However, the effect of Dex, an immunosuppressive drug, on both lung lesions and the development of MFALCs in Lpr mice has not been identified yet. Therefore, in this study, we compared the development of lung lesions and MFALCs in female Lpr mice that received either saline (saline group “SG”) or dexamethasone (dexamethasone group “DG”) in drinking water as a daily dose along with weekly intraperitoneal injections for 10 weeks. Compared to the SG group, the DG group showed a significant reduction in the levels of serum anti-dsDNA antibodies, the size of MFALCs, the degree of lung injury, the area of high endothelial venules (HEVs), and the number of proliferating and immune cells in both MFALCs and the lungs. A significant positive correlation was observed between the size of MFALCs and the cellular aggregation in the lungs of Lpr mice. Therefore, this study confirmed the ameliorative effect of Dex on the development of lung injury and MFALCs via their regressive effect on both immune cells’ proliferative activity and the development of HEVs. Furthermore, the reprogramming of MFALCs by targeting immune cells and HEVs may provide a therapeutic strategy for autoimmune-disease-associated lung injury.

## 1. Introduction

The term autoimmunity represents a condition wherein the body’s natural immune defense system cannot differentiate between its own healthy tissues and potentially harmful or foreign antigens. This causes the body to mistakenly generate an immune response against its own host cells [1]. More than 80 types of autoimmune diseases have been reported in humans, and they are found to affect different body organs, including rheumatoid arthritis (RA) in the joints [2]; inflammatory bowel arthritis (Crohn’s disease and ulcerative colitis) in the digestive tract [3]; dermatomyositis in the skin and muscles [4]; and systemic lupus erythematosus (SLE) in the joints, kidneys, heart, and skin [5]. Interestingly, pulmonary involvement, including interstitial lung disease and interstitial fibrosis, has also been reported in many autoimmune diseases, such as systemic sclerosis, SLE, and RA. These are referred to as autoimmune lung diseases [6].

Autoimmune diseases have been reported to affect both human beings (8% of the population) [7] and domestic animal species [8]. Several murine autoimmune mouse models have been successfully used to study the pathogenicity of the disease in humans. The MRL/MpJ-*Fas ^lpr/lpr^* (Lpr) mice are one of the autoimmune disease mouse models used frequently as an SLE model. They are widely used because they exhibit lesions that closely mimic the lesions found in humans. This includes lymphoproliferation, splenomegaly, lymphadenopathy, and injured lungs and kidneys. The autoimmunity is due to the increased titer of anti-double-strand DNA (dsDNA) autoantibodies in the blood [9,10]. Furthermore, we previously demonstrated a significant positive correlation between the development of lung injury and the size of the novel immunological cluster associated with mediastinal adipose tissue, which was termed mediastinal fat-associated lymphoid clusters “MFALCs” in Lpr mice [11].

Leukocyte recruitment in the lung plays a major role in the pathogenesis of lung injury. Such recruitment mainly depends on the function of chemokines and their receptors [12]. Interestingly, both chemokine (C-X-C motif) ligand 9 (CXCL9) and CXCL13 are linked to the development of autoimmune inflammatory disorders, particularly during incidences of lung injury [13,14].

Dexamethasone is one of the corticosteroids that exhibits anti-inflammatory and immunosuppressant effects [15]. Recently, the therapeutic impact of dexamethasone on the degree of lung damage associated with COVID-19 has also been reported [16]. Additionally, we have previously revealed its therapeutic effect on lupus nephritis in the Lpr autoimmune disease mouse model [17]. However, the effect of dexamethasone on the degree of lung injury and the development of MFALCs in Lpr mice remains undefined.

In the prevalence of autoimmune diseases, sex-related differences are reported in human patients, with more incidence observed in females [7]. Additionally, our previous report revealed increased development of MFALCs and lung injury in female Lpr mice compared to that of male mice [18]. In this study, we investigated the effect of dexamethasone on the lymphoproliferation of MFALCs and spontaneous immune cell infiltration within the lungs of female Lpr mice. Our results highlighted the potential ameliorative effects of dexamethasone on the development of lung injury and MFALCs. Additionally, we suggest that the underlying mechanism of such therapeutic impact may be via their regressive effect on both the immune cells’ proliferative activity and the development of high endothelial venules (HEVs), along with the suppression of chemokines.

## 2. Results

### 2.1. Analysis of the Impact of Dexamethasone on the Degree of Lung Injury and the Development of MFALCs and the Systemic Autoimmunity

#### 2.1.1. Histopathological Features of the Lungs and MFALCs in the Saline and Dexamethasone Groups

Our previous reports revealed the role of MFALCs in the development of lung injury in the Lpr autoimmune disease mice model [11,18]. However, the impact of immunosuppressive drugs on both lung lesions and the development of MFALCs in Lpr mice has not been identified yet. Therefore, in the current study, we examined sections of lung and mediastinal fat tissues (MFTs) stained with hematoxylin and eosin (HE) in both saline and dexamethasone groups. The lung tissue in the saline group showed thickening of the interalveolar septa and numerous peribronchial and perivascular mononuclear cell aggregations (MNCA) that replaced normal lung architecture with numerous collapsed alveoli. In contrast, the lung tissue of the dexamethasone group mice showed alleviation in the degree of MNCA, indicating a restoration of normal lung structure (Figure 1A). Interestingly, compared to the saline group, the morphometric measurement in the dexamethasone group revealed a significant decrease in the ratio of the area of MNCA/area of the lung field, suggesting ameliorative effects of dexamethasone on lung injury (Figure 1C).

Simultaneously, we investigated the MFTs in both study groups. Notably, the saline group showed numerous and prominent lymphoid clusters (LCs) within the MFTs, which were associated with few adipose tissues. Controversially, the dexamethasone group showed fewer and smaller size LCs but with the association of numerous adipose tissues (Figure 1B). Similarly, we also performed a histoplanimetrical analysis, where the ratio of the area of LCs/total area of MFTs was used as an index to determine the degree of the development of MFALCs between the saline and dexamethasone groups. Remarkably, a significant decrease was observed in the percentage of this ratio in the latter group compared to the former (Figure 1D).

#### 2.1.2. The Degree of Systemic Autoimmunity in the Saline and Dexamethasone Groups

Degree of lung injury and MFALC development is significantly associated with systemic autoimmunity [18]. Therefore, to examine the impact of dexamethasone treatment on systemic autoimmunity, we evaluated the titer of anti-double-stranded DNA autoantibodies (anti-dsDNA abs) in the sera of mice from both the saline and dexamethasone groups. Intriguingly, the dexamethasone group showed a significantly lower level of anti-dsDNA abs compared to that of the saline group (Figure 1E).

### 2.2. Immune Cells in the Lung and MFALCs of Saline and Dexamethasone Groups

Previously, we characterized the immune cell populations in the lung and MFALCs of Lpr mice and revealed that both lymphocytes and macrophages are predominant populations [18]. To further investigate the impact of dexamethasone on immune cell populations in both the lungs and MFALCs, the morphometrical changes were compared among the studied groups. As shown in Figure 2 and Appendix A, immunohistochemical staining and morphometrical analysis of the positive index ratios (the positive area of the immune cells/field area) were used to compare the composition of immune cells (CD3^+^ T-lymphocytes, B220^+^ B-lymphocytes, Iba1^+^ macrophages, and Gr1^+^ granulocytes) in the lungs and MFALCs between the saline and dexamethasone groups. Except for the lung granulocytes, the dexamethasone group showed decreased values for the percentages of positive immune cell index ratios in both the lungs and MFALCs compared to the saline group. No significant difference was observed in the percentages of T-lymphocytes positive index ratios in either the lungs or MFALCs between the study groups (CD3^+^ T-lymphocytes, Appendix A). However, a highly significant decrease was observed in the B-lymphocyte values in the dexamethasone group compared to the saline group (B220^+^ B-lymphocytes, Appendix A). In the case of macrophage positive area ratios, non-significant and significant differences were observed in the lungs and MFALCs, respectively (Iba-1^+^ macrophages, Appendix A). Moreover, for granulocyte positive area ratios, a highly significant increase and decrease were observed in the lungs and MFALCs, respectively (Gr-1^+^ granulocytes, Appendix A).

To characterize the Iba-1^+^ macrophage subsets in the lung tissue, triple immunofluorescence staining was performed with Iba-1 (macrophage marker), CD68 (M1 macrophage marker), and CD204 (M2 macrophage marker) antibodies. Interestingly, the examination of immunofluorescent-stained lung sections showed a decrease in the Iba-1^+^ macrophages and CD68^+^ M1 macrophages and an increased expression of the CD204^+^ M2 macrophages in the dexamethasone group compared to that of the saline group (Appendix A).

### 2.3. Analysis of the Degree of Proliferation of Immune Cells in the Lungs and MFALCs between the Saline and Dexamethasone Groups

Immunohistochemical staining of Ki67 (a proliferation marker) was performed in the lungs and MFTs of both study groups to evaluate the degree of proliferation of immune cells, which is a major characteristic of Lpr mice. Compared to the saline group, the dexamethasone group showed fewer Ki67^+^ proliferating cells in the lungs (Figure 3A) and MFALCs (Figure 3C). Consistent with our histological observations, the morphometrical analysis revealed a highly significant reduction in the percentages of the ratio of Ki67^+^ proliferating cell density (number/field area) in the lungs (Figure 3B) and the ratio of Ki67^+^ proliferating cell number/total immune cell numbers in the MFALCs (Figure 3D) in the dexamethasone group compared to the saline group.

To address the proliferating immune cell types, dual immunofluorescent staining of the lung sections was performed. The lung sections of both studied groups were co-stained with the anti-Ki67 antibody along with immune cell markers (anti-B220 Appendix A, anti-Iba1 Appendix A, and anti-Gr1 Appendix A antibodies). Interestingly, high and low numbers of Ki67^+^ B220 ^+^ B-lymphocytes were observed in the lungs of the saline and dexamethasone groups, respectively (Appendix A). However, few Ki67^+^ Iba1^+^ macrophages were observed in the lungs of both studied groups (Appendix A). On the other hand, low and high numbers of Ki67^+^ Gr1 ^+^ granulocytes were observed in the lungs of the saline and dexamethasone groups, respectively (Appendix A).

### 2.4. Comparison of HEVs and the Development of Lymphatic Vessels (LVs) in the Lung and MFALCs between the Saline and Dexamethasone Groups

Our previous reports revealed the role of both HEVs and LVs in the development of MFALCs and lung injury [19,20]. To investigate the degree of HEVs and the development of LVs in the lungs and MFTs, immunostaining was performed using peripheral node addressin (PNAd) and lymphatic vessel endothelial hyaluronic acid receptor-1 (LYVE-1) antibodies, respectively. Interestingly, the dexamethasone group showed fewer prominent HEVs in the lungs (Figure 4A) and MFALCs (Figure 4B) compared to those of the saline group. Next, we performed a morphometrical analysis of the percentage of HEV relative area ratios (area of the HEVs/area of the field) in the lungs (Figure 4C) and MFALCs (Figure 4D). Importantly, statistical analysis revealed a highly significant reduction in this ratio in the dexamethasone group compared to that of the saline group in both lungs and MFALCs (Figure 4C,D).

For the examination of the degree of LV development in the lung, dual immunofluorescent staining for LYVE-1/CD68 was performed. Interestingly, more numerous LYVE-1^+^ LVs and CD68^+^ macrophages were observed in the lung sections of the saline group than in those of the dexamethasone group. Furthermore, most of the LVs in the lungs of the saline group, but not of the dexamethasone group, were engorged with immune cells (Figure 5A), and their lining epithelial showed a positive reaction to Ki67 immunostaining (Appendix A). Simultaneously, morphometrical analysis of the LYVE-1^+^ LVs relative area ratio was performed for the LYVE-1 immunohistochemical-stained lung and MFALC sections. The dexamethasone group mice showed a highly significant reduction in this ratio in both lungs and MFALCs in comparison to that of the saline group mice (Figure 5B,C).

### 2.5. Analysis of the Impact of Dexamethasone on Chemokine Expression in the Lungs between the Saline and Dexamethasone Groups

We examined the expression of CXCL9, CXCR5, and CXCL13 in the lungs of both study groups. Interestingly, CXCL9 was previously reported to induce the chemotaxis, differentiation, and proliferation of leukocytes, especially CD8 and CD4 positive T lymphocytes [21,22]. Therefore, double immunofluorescence was performed to detect CXCL9^+^ cells along with either CD4^+^ or CD8^+^ T lymphocytes in the lung sections of both study groups. As shown in Appendix A, numerous CXCL9^+^, CD4^+^, and CD8^+^ cells were observed in the lungs of the saline group compared to those of the dexamethasone group. Of note, many CXCL9^+^ cells were mainly observed within the area of MNCA in the lungs of the saline group mice. Moreover, numerous CXCL9^+^ cells showed co-staining with the CD8^+^ T lymphocytes (Appendix A) but not with CD4^+^ T lymphocytes (Appendix A) in the lungs of the saline group rather than the dexamethasone group.

Importantly, the CXCR5 receptor and its ligand (CXCL13) have been reported to be involved in B cell-selective chemotaxis and recruitment [23,24]. Thus, the co-expression of either CXCR5/CD79a positive B lymphocytes or CXCL13 and B220 positive B lymphocytes was examined in the lungs of both study groups. As shown in Figure 6A, more positive CXCR5 and CD79a cells were observed in the lungs of the saline group than in those of the dexamethasone group. Interestingly, the lungs of the saline group, but not the dexamethasone group, revealed numerous CXCR5^+*/*^CD79a^+^ B lymphocytes co-stained cells. Additionally, more positive CXCL13 and B220 cells were observed in the saline group compared to the dexamethasone group. In the saline group, numerous B220^+^ B-lymphocytes were mainly observed within the MNCA. However, the CXCL13^+^ cells were mainly observed within the vicinity of the MNCA (Figure 6B). Furthermore, a significantly higher percentage for the ratio of CXCR5^+^ and CXCL13^+^ cell densities (number/mm^2^ lung area) was observed in the saline group compared to that of the dexamethasone group (Figure 6C,D).

### 2.6. Analysis of Histopathological Correlations among Different Parameters of the Lungs and MFALCs in Both the Saline and Dexamethasone Groups

Figure 7A,B shows significant positive correlations among the size of MFALCs, the lung MNCA area ratio (Figure 7A), and the HEV area ratio in the MFALCs (Figure 7B). Interestingly, the correlations among the HEV area ratio, MFALC HEVs (Figure 7C), and MNCA in the lungs (Figure 7D) also show a highly significant positive correlation (Figure 6C,D). This suggests the possible role of HEVs in the chemoattraction of the immune cells of MFALCs to the lungs and also the development of numerous MNCA in the lungs of the saline group mice.

### 2.7. Analysis of Histopathological Correlations between Immune Cell Populations and the Proliferating of Cells in the Lungs and MFALCs of Both Saline and Dexamethasone Group Mice

The correlations between the immune cell populations (B220^+^ B-lymphocytes, CD3^+^ T-lymphocytes, Iba1^+^ macrophages, and Gr-1^+^ granulocytes) and the degree of proliferation of the cells in the lungs and MFALCs were analyzed in both the study groups (Figure 8). Interestingly, the B-lymphocyte population in the lungs showed a highly significant positive correlation with MFALCs (Figure 8A). Furthermore, a significant and positive correlation was observed in the proliferation of cells among the lungs and MFALCs of both the study groups (Figure 8E). However, the T-lymphocyte (Figure 8B) and macrophage (Figure 8C) populations showed a non-significant positive correlation between the lungs and MFALCs, while a non-significant negative correlation was observed between the lungs and MFALCs granulocytes (Figure 8D).

## 3. Discussion

The present study examined the impact of dexamethasone, an anti-inflammatory corticosteroid, on the degree of lung injury and the development of MFALCs in the Lpr autoimmune mouse model. This mouse model was previously reported to develop severe lung injury and prominent MFALCs associated with autoimmunity [11]. Corticosteroids, especially dexamethasone, have been successfully used in the treatment of patients with acute lung injury, COVID-19, and idiopathic pulmonary fibrosis [25,26,27]. Additionally, our recent study revealed that the administration of dexamethasone in the Lupus nephritis Lpr autoimmune mice model led to the alleviation of renal lesions via the reduction in vasculature-associated lymphoid tissues in the kidney of Lpr mice [17]. However, no clarifications are available on the effect of dexamethasone on the degree of lung injury and the development of MFALCs. Moreover, the correlations between MFALC development and the amelioration of lung injury after the administration of dexamethasone have not been clarified yet. Therefore, we aimed to clarify the possible mechanisms through which dexamethasone exerts its ameliorative effect. We examined the lungs and MFTs of both saline- and dexamethasone-administered Lpr mice. Interestingly, our previous study revealed sex-related differences among the Lpr mice in the presentation of the degree of lung injury and development of MFALCs, with the number of lung lesions and prominent MFALCs being greater in female mice [18]. Furthermore, women are reported to be more susceptible to autoimmune diseases compared to men [1]. Therefore, instead of male mice, we used female Lpr mice in this study.

Our results showed that, compared to the saline group, the lungs of the dexamethasone group mice showed less MNCA along with the restoration of normal alveolar morphology, which suggests an ameliorative effect of dexamethasone on the severity of lung injury. This was also in line with the results presented by Villar et al. [28], who reported the efficacy of dexamethasone in the treatment of acute respiratory distress syndrome in patients with COVID-19. Next, we also confirmed that the dexamethasone-administered group showed a significant decrease in the size and number of MFALCs. Our results revealed a significant positive correlation between the degree of lung injury and the development of MFALCs, which strongly suggested a major role for MFALCs in the development of lung injury. Notably, the dexamethasone group showed a significant decrease in the serum anti-dsDNA abs (as an autoimmune index) compared to that of the saline group. Therefore, we postulated that the administration of dexamethasone could control the development of MFALCs and lung injury via a systemic effect.

Also, HEVs and LVs play a major role in the lymphoid organs and tissues. HEVs are specialized postcapillary venules composed of cuboidal blood endothelial cells. These play a role in chemoattracting naïve and central memory cells. Furthermore, the role of LVs has been reported to include transporting antigens, lymphocytes, and antigen-presenting cells in and out of the lymph nodes [29]. Importantly, our previous study revealed a major role for both HEVs and LVs in the development of both lung injury and MFALCs in several mice models [11,19,20]. Predictably, our present investigation revealed less developed HEVs and LVs in both the lungs and MFALCs of the dexamethasone-administered group compared to those of the saline group. Moreover, significant positive correlations were observed between the MNCA and the lung HEVs, as well as between the size of MFALCs and their HEVs. Additionally, since our results revealed a positive and highly significant correlation between the development of HEVs both in the lungs and MFALCs, we strongly suggest a major role for both HEVs and LVs in the development of lung injury and MFALCs. Additionally, we suggest that the ameliorative effect of dexamethasone may be induced via its interference with the development of both HEVs and LVs.

To understand the impact of dexamethasone on the intrathoracic immune hemostasis, we examined the immune cells and their degree of proliferation in both the lungs and MFALCs of dexamethasone and saline group mice. Our results revealed that, compared to the saline group, all the studied immune cell populations were alleviated, except for the lung granulocytes, in the lungs and MFALCs of the dexamethasone-administered mice. A non-significant difference was observed in the alleviation of T-lymphocytes in the lungs and in macrophages and T-lymphocyte populations within the MFALCs. However, a highly significant reduction was observed in the B-lymphocyte populations within the lungs and MFALCs of the dexamethasone group mice. Additionally, compared to the saline group, a significant reduction was observed in the CXCL13 chemokine within the lungs of the dexamethasone group mice. CXCL13 has been reported to play a major role in the pathogenesis of several autoimmune diseases, lymphoproliferative disorders, inflammatory conditions, as well as lymphoid neogenesis [30]. Moreover, this chemokine is known as B cell-attracting chemokine 1 or B lymphocyte chemoattractant, and it plays an important role in the process of selectively attracting B lymphocytes [31]. Interestingly, the lungs of the saline group mice revealed several CXCL13^+^ cells in the confines of MNCA that showed numerous B-lymphocyte populations. Therefore, we postulated that dexamethasone could induce alleviation in the B-lymphocyte population via the suppression of CXCL13. Furthermore, the lungs of the saline group revealed a significantly higher number of CXCR5^+^ cells as well as more numerous CXCR5^+^ B220^+^ B lymphocytes, especially within the MNCA, compared to the dexamethasone group. Interestingly, recent reports have focused on the role of CXCL13 and its receptor “CXCR5” as a B-cell chemoattractant, especially in autoimmune diseases [30].

The current investigation also examined the role of CXCL9 in the recruitment of CD4^+^ helper and CD8^+^ cytotoxic T lymphocytes in the lung tissue of the studied groups. More expressions for CXCL9, CD4, and CD8 were observed in the saline group lungs than those of the dexamethasone group. Furthermore, our results revealed the co-expression of CXCL9 along with CD8^+^ cytotoxic T lymphocytes in the lungs of the saline group in contrast with the dexamethasone group. Similarly, the role of CXCL9 in CD8^+^ cytotoxic T lymphocytes has been recently clarified [32].

Several reports have clarified the role of macrophage polarization in the pathogenesis and resolution of lung injury [33]. There are two main macrophage subsets that are involved in the pathogenesis of inflammation progression or recovery: the classically activated (or inflammatory) M1 macrophages and the alternatively activated M2 (or anti-inflammatory) macrophages [34]. Functionally, the M1 macrophages are mainly involved in pro-inflammatory activities, chemotaxis, and antimicrobial activities [35]. On the contrary, the M2 macrophages are engaged in the suppression of effector T cells and the promotion of tissue remodeling. Hence, they are beneficial for the resolution of inflammation and tissue repair [36]. This was also in line with our observation of macrophage subsets in the lungs of the saline group mice, where the saline group showed more and less numerous M1 and M2 macrophages, respectively, compared to those of the dexamethasone group. This strongly suggested that the polarization from M1 to M2 macrophages could play an important role in the resolution of lung injury following the administration of dexamethasone.

Surprisingly, our results revealed that, compared to the saline group, the dexamethasone group mice showed a significant increase in the granulocyte populations in the lungs. Hence, the recruitment of granulocytes, especially neutrophils, to the lungs has been considered as perhaps playing a key role in the pathogenesis of the progression of lung injury and targeting granulocyte recruitment has recently been considered as a therapeutic strategy [37,38]. At first glance, these data seem to contradict our observations. However, other studies have provided evidence that, after the occurrence of inflammatory conditions, the neutrophil phenotype can be changed from a pro-inflammatory type to an anti-inflammatory type through the production of pro-resolving mediators, such as lipoxins, after which inflammation is resolved [39].

Importantly, our findings revealed a negative correlation among the granulocyte populations but positive correlations for other studied immune cell populations (macrophages and T- and B-lymphocytes) within the lungs and MFALCs. This suggests that the latter immune cell types within the MFALCs may have a role in the pathogenesis of lung injury rather than the granulocytes. Moreover, our results revealed a significant reduction in the degree of the proliferation of immune cells following dexamethasone administration, which suggested another postulating ameliorative effect of dexamethasone on the degree of lung injury in the Lpr mice characterized by lymphoproliferative disorder [40]. Interestingly, our results also revealed significant alleviation in the expression of lung CXCL9 in the dexamethasone group compared to that of the saline group. CXCL9 plays an essential role in the induction of the chemotaxis, differentiation, and proliferation of leukocytes [21]. Therefore, we postulated that the impact of dexamethasone on resolving immune cell proliferation from an injured lung could be via the reduction in the expression of CXCL9 and its proliferation tendency.

## 4. Materials and Methods

### 4.1. Experimental Animals and Ethical Statement

Female Lpr mice (aged 12 weeks) were purchased from Japan SLC Inc. (Shizuoka, Japan) and maintained in specific pathogen-free housing conditions. The researchers adhered to the Ethical Guidelines for Laboratory Animal Care and Use, which was approved by the Institutional Animal Care and Use Committee of the Graduate School of Veterinary Medicine, Hokkaido University (Approval No. 15-0079).

### 4.2. Experimental Design and Sample Preparation

Administration of dexamethasone was performed according to our previous study [17]. Briefly, the Lpr mice were divided into the following two groups: the dexamethasone group and the saline group. The former group received dexamethasone with drinking water as a daily oral dose along with weekly intraperitoneal injections (for 10 weeks) at a dose of 0.4 mg/kg of body weight. The same protocol was followed for the latter group, but the dexamethasone was replaced with saline. At 22 weeks of age, deep anesthesia was administered to all the mice of both groups (intraperitoneal injection of 0.2 mL anesthetic solution “mixture of 5.0 mg/kg B.W. butorphanol, 4.0 mg/kg B.W. midazolam, and 0.3 mg/kg body weight medetomidine”/10 gm body weight). Next, the blood was collected from the femoral artery. Later, the head of the mouse was dislocated, and both the lung and mediastinal fat tissues (MFTs) were immediately collected and immersed in 4% paraformaldehyde for overnight fixation at 4 °C. The tissues were then processed for paraffin embedding.

### 4.3. Serum Autoantibody Measurement

To examine the severity of systemic autoimmune disease, the levels of serum anti-dsDNA abs were used as the index of autoimmunity, this was measured using an ELISA kit (FUJIFILM Wako Pure Chemical Corporation, Osaka, Japan) as per the manufacturer’s instructions.

### 4.4. Tissue Preparation for Histopathological and Immunohistochemical Analysis

Paraffin sections of 3 µm were prepared from MFT and lung tissues. These sections were subjected to either hematoxylin and eosin (HE) staining for routine histopathological examination or immunostaining according to our previous study [11].

Immunohistochemical staining was performed to detect the immune cells (B220+ B-lymphocytes, CD3^+^ T-lymphocytes, Iba1^+^ macrophages [41,42,43], and Gr1^+^ granulocytes), proliferating cells (Ki67), and the vessels (peripheral node addressin “PNAd”, high endothelial venules “HEVs”, and lymphatic vessel endothelial hyaluronic acid receptor 1 “LYVE-1” lymphatic vessels “LVs”). After deparaffinization and rehydration, heat-induced antigen retrieval was applied to the paraffin sections. The sections were incubated in absolute methanol (containing 3% hydrogen peroxide) for 20 min at room temperature to quench the endogenous peroxidase activity. This was followed by flushing the sections with distilled water (DW). Then, the sections were incubated with normal serum for 1 h at room temperature to block the non-specific binding sites. After blocking, the sections were incubated overnight with specific primary antibodies at 4 °C (Table 1). Next, the sections were rinsed with phosphate buffer saline (PBS) 3 times, with each rinse performed for 5 min. The washed sections were incubated with specific biotinylated secondary antibodies for 30 min at room temperature. Later, the sections were rinsed with PBS (3 times/5 min each) and incubated with streptavidin for 30 min at room temperature. The sections were again washed with PBS, and the positive reactions were developed by adding 3, 3′ diaminobenzidine tetrahydrochloride-H_2_O_2_ solution for a short incubation time. Next, the nuclei were counterstained with hematoxylin. Finally, the sections were dehydrated, cover slipped, and examined under a microscope.

Additionally, immunofluorescent staining of lung sections was performed to detect the expression of LVs along with macrophages, chemokine ligands (CXCL9, CXCR5, and CXCL13), along with either T or B- lymphocyte markers and Iba1^+^ macrophage subsets (CD68^+^ M1 and CD204^+^ M2 macrophages [44]). Briefly, the lung sections from both the study groups were deparaffinized and rehydrated and then subjected to antigen retrieval by heating (Table 1), followed by blocking with normal donkey serum for one hour in a humid chamber. Next, the sections were washed with PBS (3 times/5 min each) and incubated overnight with a mixture of primary antibodies. Double immunofluorescent staining was performed for detection of LVs/macrophages (using rabbit anti- LYVE-1/rat anti- anti-CD68); proliferating immune cells (using rabbit anti-Ki67/rat anti-B220 (FITC); rabbit anti-Ki67/goat anti-Iba-1; and rabbit anti-Ki67/rat anti-Gr-1); and chemokine expression (using goat anti-CXCL9/rat anti-CD4; goat anti-CXCL9/rat anti-CD8; rat anti-CXCR5/mouse anti- CD79a “B-lymphocyte cell marker [45]” and goat anti-CXCL13/rat anti-B220). Triple immunofluorescent staining was performed for detection of macrophage subsets (using goat anti-Iba1/mouse anti-CD204/rabbit anti-CD68 antibodies). The sections were then washed with PBS and incubated for 30 min with a mixture of the corresponding secondary antibodies (Alexa-Fluor-546-labeled donkey anti-mouse IgG, Alexa-Fluor-546-labeled donkey anti-rabbit IgG, Alexa-Fluor-647-labeled donkey anti-goat IgG, Alexa-Fluor-488-labeled donkey anti-rabbit IgG, and Alexa-Fluor-488-labeled donkey anti-rat IgG) (Invitrogen, Eugene, OR, USA) at a dilution of 1:500 each. For nuclear staining, the sections were incubated with Hoechst 33,342 (at dilution 1:2000) (Dojindo, Kumamoto, Japan) for 2 min, and the sections were mounted for observation. Finally, fluorescence microscopy was used for observing the sections (Model BZ-X710 microscope) (Keyence, Osaka, Japan). Table 1 lists the details of antibodies, working dilution, and antigen retrieval conditions.

### 4.5. Morphometrical Measurements

NanoZoomer 2.0-RS (Hamamatsu Photonics, Shizuoka, Japan) was used to prepare the virtual slides of HE-stained lung and MFT sections to analyze the percentages of the ratios for the area of the MNCA/area of the lung field and the area of LCs/the area of total MFTs between the study groups. The NDP view2 software (Hamamatsu Photonics Co., Ltd., Hamamatsu City, Japan) was used to measure the area of MNCA/area of the lung field and the area of the LCs/area of the total MFTs, after which the percentages of the averages were compared between the study groups. Similarly, the NDP view2 software was also used to measure the ratios of HEVs and the area of LVs/both the total lung field and the area of MFALCs in the virtual slides, converted to immuno-stained PNAd and LYVE-1 lung and MFT sections, which were then compared between both groups.

To measure the positive index ratio of the immune cells (CD3^+^ T-lymphocytes, B220^+^ B-lymphocytes, Iba1^+^ macrophages, and Gr1^+^ granulocytes), we captured digital images of immunohistochemically stained lungs and MFTs at 20× magnification using the BZ-X710 microscope. Next, we measured both the field area and immune cell positive area within the corresponding field area using a BX-analyzer (Keyence, Osaka, Japan). Then, the average ratios of immune cells positive area/field area were calculated and compared between the study groups. A BX-analyzer (Keyence) was also used to quantify the degree of proliferation, the ratio of Ki67 positive cell density (number/mm^2^), and the number of Ki67^+^ cells/the total number of immune cells in the lungs and MFALCs. Similarly, the cell densities of CXCR5^+^ and CXCL13^+^ cells in the lung sections were measured, and the averages were compared between the study groups.

### 4.6. Statistical Analysis

The statistical analyses among the study groups were conducted using the Graph pad Prism 6 software (GraphPad Software, San Diego, CA, USA). The differences among different numerical results in the saline and dexamethasone groups were compared using the Mann–Whitney *U* test. The data were presented as mean ± standard error (SE). Significant (*) and highly significant (**) differences were considered at *p* values < 0.05 and <0.01, respectively. Additionally, the correlation coefficients between two different parameters were determined using Spearman’s correlation test (*p* < 0.05).

## Figures and Tables

**Figure 1 ijms-23-04449-f001:**
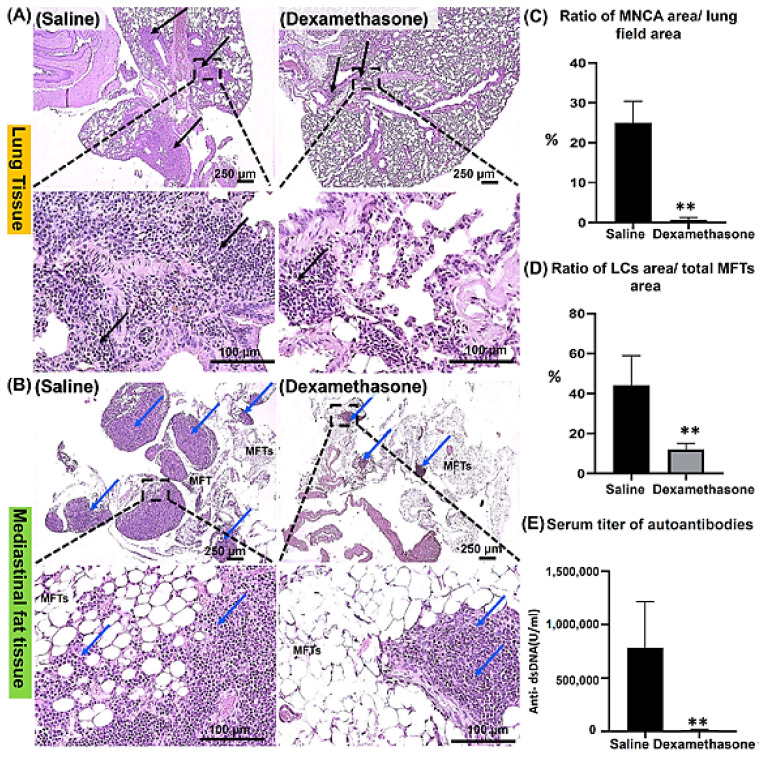
Analysis of the effect of dexamethasone on the degree of lung injury, development of mediastinal fat associated-lymphoid clusters (MFALCs), and autoimmunity in female Lpr mice. (**A**,**B**) HE staining of the lung (**A**) and mediastinal fat tissues “MFTs” (**B**) in saline and dexamethasone administered groups. Notice the mononuclear cellular aggregations “MNCA” (black arrows) and MFALCs (blue arrows) associated with the MFTs. (**C**,**D**) Graphs indicating the morphometrical data of the percentages for the ratios of the area of MNCA/area of the lung field (mm^2^) and the area of LCs/the total area of MFTs (mm^2^). (**E**) Graph indicating the serum levels of anti-double-stranded DNA autoantibodies. Highly significant values (**) were observed between the saline and dexamethasone groups (*p* < 0.01), where *n* = 5 in each experimental group. Analysis was conducted using the Mann–Whitney *U* test. Data are presented as mean values ± SE.

**Figure 2 ijms-23-04449-f002:**
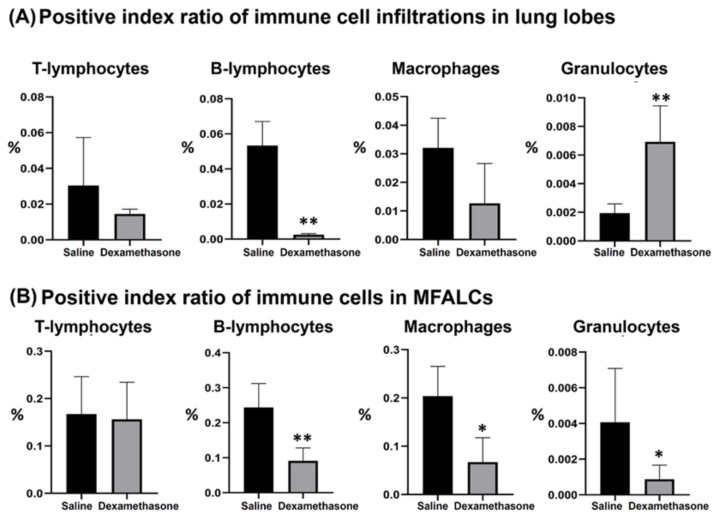
Morphometrical analysis of the effect of dexamethasone on immune cell populations in the lungs and mediastinal fat associated-lymphoid clusters (MFALCs) in female Lpr mice. (**A**,**B**) Representative graphs showing the percentage of the positive index ratios for CD3^+^ T-lymphocytes (Appendix A), B220^+^ B-lymphocytes (Appendix A), Iba-1^+^ macrophages (Appendix A), and Gr-1^+^ granulocytes (Appendix A) in the lungs (**A**) and MFALCs (**B**). Significant (*) and highly significant values (**) were observed between the saline and dexamethasone groups (*p*  values < 0.05, and <0.01, respectively), where *n* = 5 in each experimental group. Analysis was conducted using the Mann–Whitney *U* test. Data are presented as mean values ± SE.

**Figure 3 ijms-23-04449-f003:**
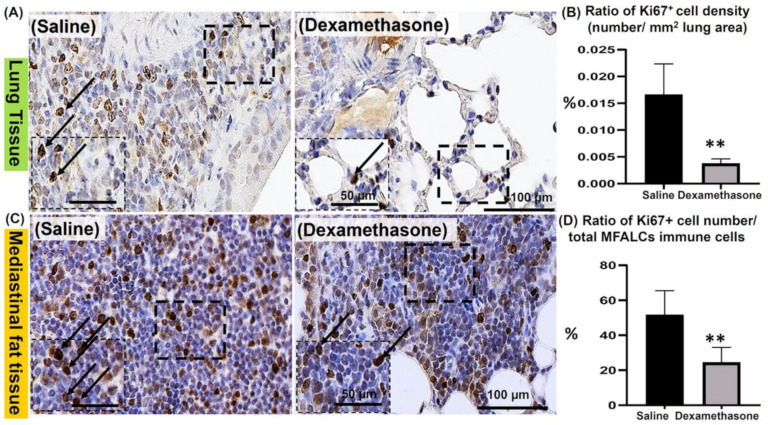
Analysis of the degree of proliferation of immune cells in the lungs and mediastinal fat associated-lymphoid clusters (MFALCs) in female Lpr mice. (**A**,**C**) Representative images of the immunohistochemical-stained lung (**A**) and mediastinal fat tissue sections (**C**) stained with anti-Ki67 antibody in both the saline and dexamethasone groups. (**B**,**D**) Representative graphs showing the percentage for the ratio of Ki67^+^ proliferating cell density in the lungs (**B**) and Ki67^+^ proliferating cell number/total immune cell number in the MFALCs (**D**). A highly significant value (**) was observed between the saline and dexamethasone groups (*p* values < 0.01), where *n* = 5 in each experimental group. Analysis was conducted using the Mann–Whitney *U* test. Data are presented as mean values ± SE.

**Figure 4 ijms-23-04449-f004:**
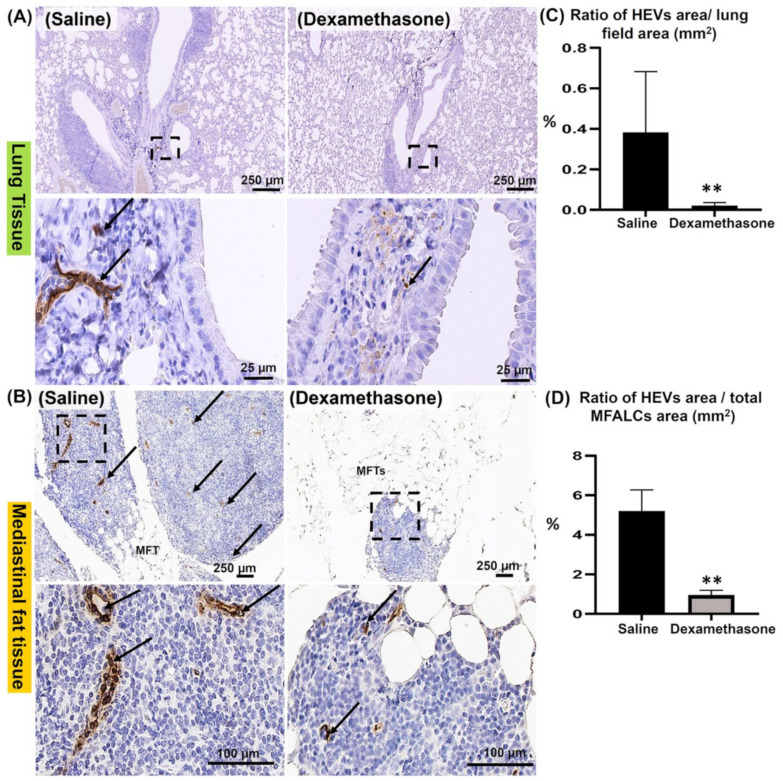
Analysis of the effect of dexamethasone on the degree of the development of HEVs in the lungs and mediastinal fat associated-lymphoid clusters (MFALCs) in female Lpr mice. (**A**,**B**) Representative images of the immunohistochemical-stained lung (**A**) and mediastinal fat tissue sections (**B**) stained with anti-PNAd antibody in both the saline and dexamethasone groups. Notice PNAd^+^ HEVs (arrows). (**C**,**D**) Representative graphs showing the percentage of the relative area ratio of PNAd^+^ HEVs in the lungs (**C**) and MFALCs (**D**). A highly significant value (**) was observed between saline and dexamethasone groups (*p* values < 0.01), where *n* = 5 in each experimental group. Analysis was conducted using the Mann–Whitney *U* test. Data are presented as mean values ± SE.

**Figure 5 ijms-23-04449-f005:**
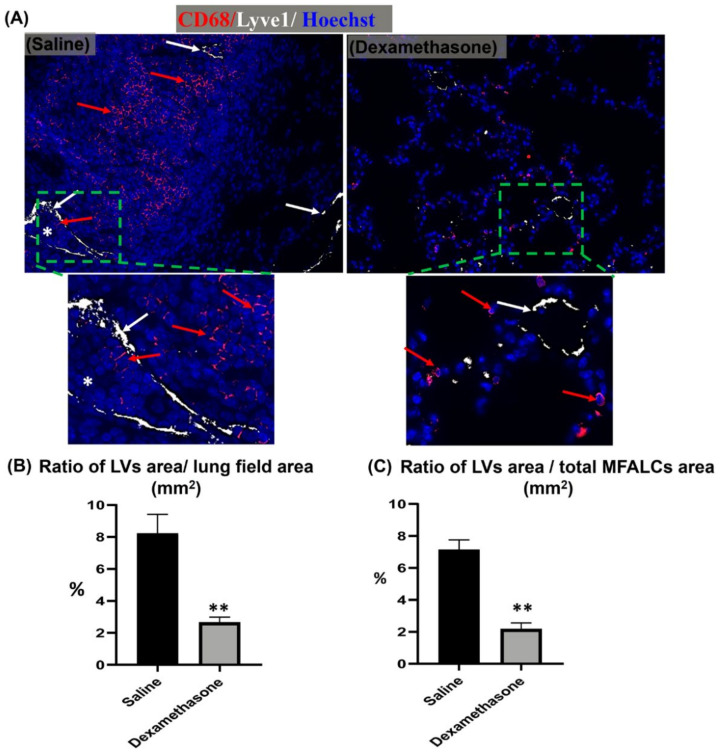
Analysis of the effect of dexamethasone on the degree of the development of lymphatic vessels (LVs) in the lungs and mediastinal fat associated-lymphoid clusters (MFALCs) in female Lpr mice. (**A**) Representative images of the dual immunofluorescent stained lung section with anti- CD68^+^ macrophages (red) and anti- LYVE-1 (white) antibodies. Notice CD68^+^ macrophages (red arrows), LYVE-1^+^ LVs (white arrows), and engorged LVs with immune cells (*). (**B**,**C**) Representative graphs showing the percentages of LYVE-1^+^ LV relative area ratios in the immunohistochemical-stained lungs (**B**) and MFALCs (**C**) tissue sections. Highly significant values (**) were observed between the saline and dexamethasone groups (*p* values < 0.01), where *n* = 5 in each experimental group. Analysis was conducted using the Mann–Whitney *U* test. Data are presented as mean values ± SE.

**Figure 6 ijms-23-04449-f006:**
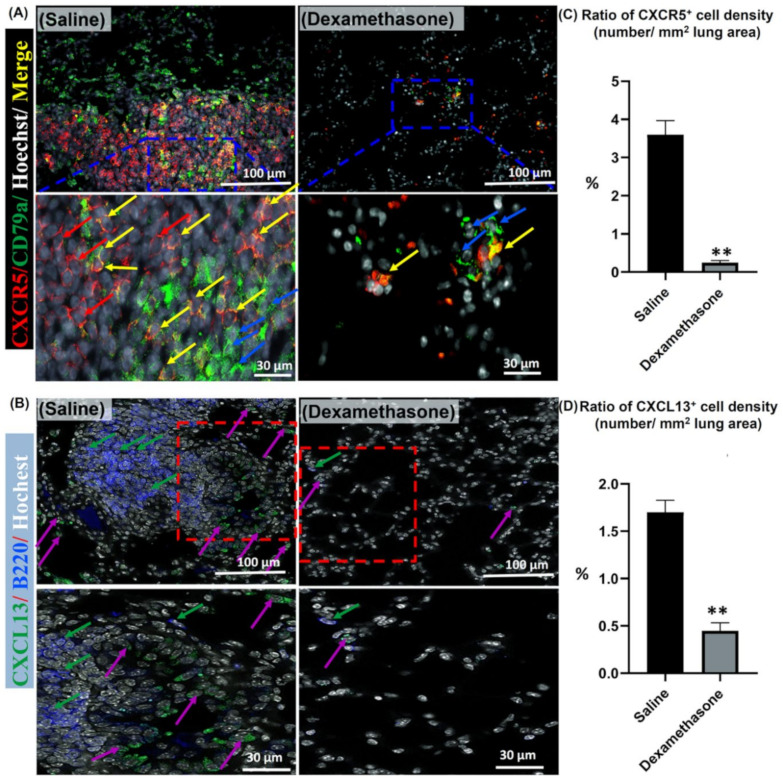
Analysis of the effect of dexamethasone on the expression of chemokine in the lungs of female Lpr mice. (**A**,**B**) Representative merged images of double immunofluorescent staining for CXCR5 “red” and CD79a “green” positive cells, and Hoechst nuclear staining “blue” (**A**) along with CXCL13 “green” and B220 “blue” positive cells, and Hoechst nuclear staining “white” (**B**). Notice positive reactions for CXCR5 (red arrows), B-lymphocytes (blue arrows), CXCL13 (violet arrows), and CXCR5^+/^CD79a^+^ co-stained cells (yellow arrows). (**C**,**D**) Representative graphs showing the percentage for the ratio of positive cell density for the CXCR5 (**C**) and CXCL13 chemokines (**D**). A highly significant value (**) was observed between saline and dexamethasone groups (*p* values < 0.01), where *n* = 5 in each experimental group. Analysis was conducted using the Mann–Whitney *U* test. Data are presented as mean values ± SE.

**Figure 7 ijms-23-04449-f007:**
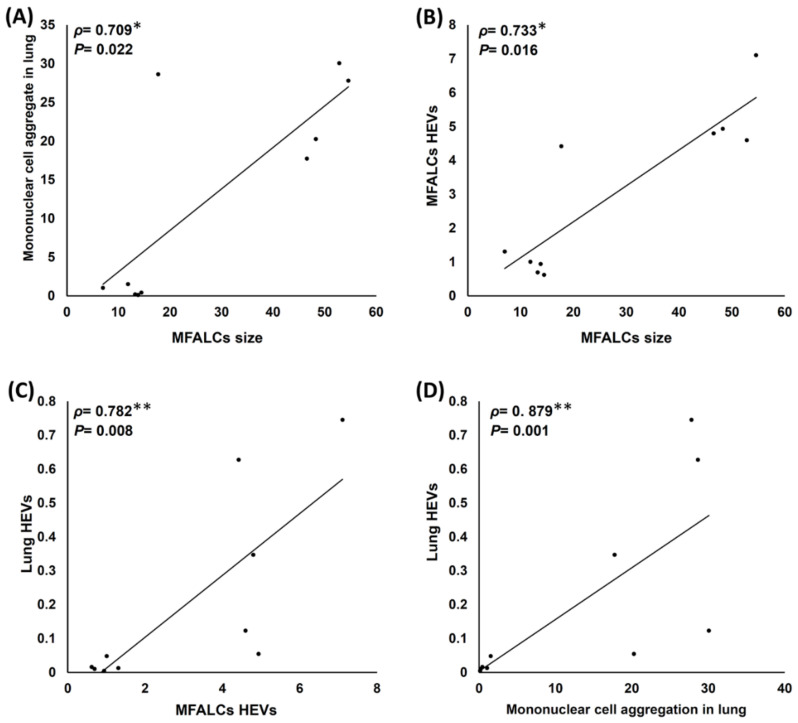
Analysis of the correlations among histoplanimetrical measurements of the lungs and mediastinal fat associated-lymphoid clusters (MFALCs) in the female Lpr mice. (**A**,**B**) Representative graphs showing Spearman’s correlations among the size of MFALCs with mononuclear cell aggregates in the lung (**A**) and HEVs in the MFALCs (**B**). (**C**,**D**) Representative graphs showing Spearman’s correlations among the lung HEVs with MFALC HEVs (**C**) and mononuclear cell aggregation in the lungs (**D**); The data were analyzed by the Spearman’s correlation test, where *n* = 10, *ρ*: Spearman’s rank-order correlation coefficient. * Significant, *p* < 0.05, ** Highly significant, *p* < 0.01.

**Figure 8 ijms-23-04449-f008:**
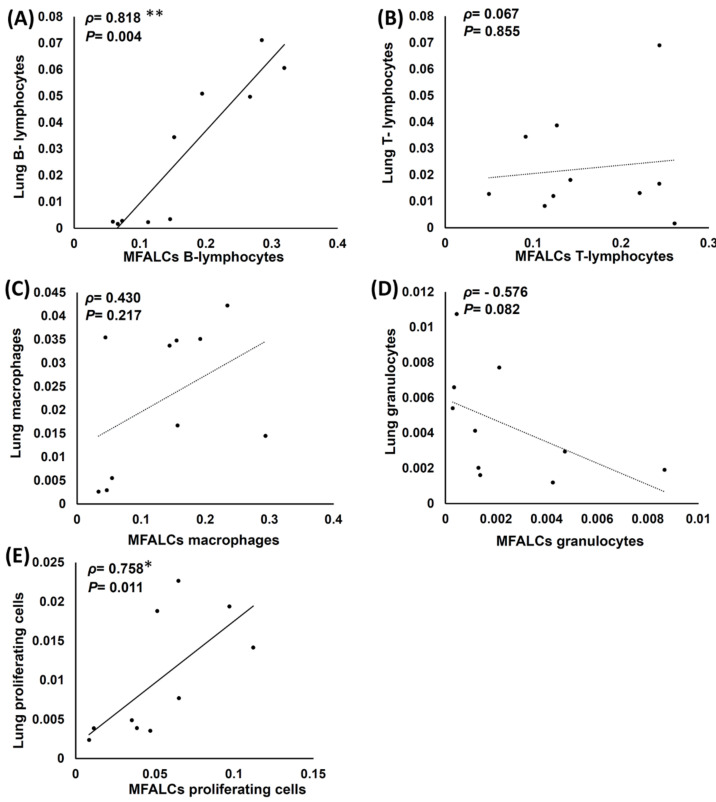
Analysis of the correlations between the immune cell populations and the proliferating cells in the lungs and mediastinal fat associated-lymphoid clusters (MFALCs) in saline and dexamethasone group mice. (**A**–**E**) Representative graphs showing Spearman’s correlation between the immune cell populations of the lungs and MFALCs (**A**), B-lymphocyte populations of the lungs and MFALCs (**B**), macrophage populations of the lungs and MFALCs (**C**), granulocyte populations of the lungs and MFALCs (**D**), proliferating cell populations of the lungs and MFALCs (**E**). The data were analyzed by Spearman’s correlation test, where *n* = 10, *ρ*: Spearman’s rank-order correlation coefficient. * Significant, *p* < 0.05, ** Highly significant, *p* < 0.01.

**Table 1 ijms-23-04449-t001:** List of antibodies, source, working dilutions, and conditions for antigen retrieval.

Antibody	Source	Dilution	Antigen Retrieval	Heating Condition
Rabbit anti-CD3	Nichirei (Tokyo, Japan)	1:200	20 mM Tris-HCl buffer (pH 9)	105 °C, 20 min
Rat anti-B220	Cedarlane (Ontario, Canada)	1:1600	10 mM citrate buffer (pH 6.0)	105 °C, 20 min
Rabbit anti-Iba1	Wako (Osaka, Japan)	1:1200	10 mM citrate buffer (pH 6.0)	105 °C, 20 min
Rat anti-Gr1	R and D system (Minneapolis, MN, USA)	1:800	0.1% pepsin/0.2 N HCl	37 °C, 5 min
Rabbit anti-Ki67	Abcam (Tokyo, Japan)	1:800	20 mM Tris-HCl buffer (pH 9)	105 °C, 20 min
Rabbit anti-LYVE-1	Adipogen (San Diego, CA, USA)	1:500	10 mM citrate buffer (pH 6.0)	105 °C, 20 min
Rat anti-PNAd	BioLegend (San Diego, CA, USA)	1:500	20 mM Tris-HCl buffer (pH 9)	105 °C, 20 min
Goat anti-CXCL9	R and D system (Minneapolis, MN, USA)	1:400	20 mM Tris-HCl buffer (pH 9)	105 °C, 20 min
Goat anti-CXCL13	R and D system (Minneapolis, MN, USA)	1:200	20 mM Tris-HCl buffer (pH 9)	105 °C, 20 min
Goat anti-Iba1	Abcam (Tokyo, Japan)	1:600	20 mM Tris-HCl buffer (pH 9)	105 °C, 20 min
Rabbit anti-CD68	Abcam (Cambridge, UK)	1:600	20 mM Tris-HCl buffer (pH 9)	105 °C, 20 min
Mouse anti-CD204	TransGenic (Hyogo, Japan)	1:400	20 mM Tris-HCl buffer (pH 9)	105 °C, 20 min
Rat anti-CD68	Biolegend (San Diego, CA, USA)	1:100	10 mM citrate buffer (pH 6.0)	105 °C, 20 min
Rat anti-B220 (FITC)	Southern Biothech (Birmingham, AL, USA)	1:300	20 mM Tris-HCl buffer (pH 9)	105 °C, 20 min
Rat anti-CD4	Southern Biothech (Birmingham, AL, USA)	1:200	20 mM Tris-HCl buffer (pH 9)	105 °C, 20 min
Rat anti-CD8	Southern Biothech (Birmingham, AL, USA)	1:400	20 mM Tris-HCl buffer (pH 9)	105 °C, 20 min
Mouse anti-CD79a	Novus Biologicals (Centennial, CO, USA)	1:500	10 mM citrate buffer (pH 6.0)	90 °C, 30 min
Rat anti-CXCR5	BD Pharmingen (Tokyo, Japan)	1:100	10 mM citrate buffer (pH 6.0)	90 °C, 30 min

## Data Availability

The data presented in this study are available within the article text and figures.

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
