# Peer review of "The Ameliorative Effect of Dexamethasone on the Development of Autoimmune Lung Injury and Mediastinal Fat-Associated Lymphoid Clusters in an Autoimmune Disease Mouse Model"

_ijms, 2022, doi:10.3390/ijms23084449_

Round 1
Reviewer 1 Report
In this manuscript, Elewa et al. a mouse model (Lpr) of systemic lupus erythematosus (SLE), to examine the effect of dexamethasone on histological features in the affected lung and MFALCs (mediastinal fat-associated lymphoid clusters), including lung/lymphoid tissue injury, immune infiltration, and vasculature.
My biggest concern is how this study will elucidate impactful mechanisms of SLE or novel alternative therapeutic strategies for SLE treatment. Moreover, the figure arrangement and logical flow feel like showing us a check list of scientific findings rather than a scientific story. In many figures little rationale was provided to tell readers which experiments were to address what scientific questions, and how they fit into a bigger picture.
I also have listed below some specific concerns:
CD68 is a pan-macrophage/monocyte marker, not specific to M1s.
Iba1 is not a lung macrophage marker. On one hand it can be expressed on DCs; on the other hand many lung macrophages, especially CD11b-negative resident macrophages do not express Iba1.
The degree of proliferation and interpretation is confusing. To address what cells are proliferating, dual staining is necessary. It is unclear if Figure 3D was quantified by calculating the ratio of Ki67 positivity over immune cell percentage (which is not an appropriate analysis), or quantifying the number of Ki67-expressing leukocytes (if so, example of staining should be shown).
Lyve1 is not only on lymphatic endothelium, but also on macrophages, especially M2s. Co-staining with CD31 or CD68 is needed.
It is unclear what was the purpose to assess Lyve1. Recruitment of effector leukocytes to the lung relies on blood vessels, not lymphatic vessels. Blood vessels are the logical candidate of interest here.
Along the same line, it is unclear how co-staining of CXCL9 with Lyve1 is informative. CXCL9 attracts CD8 and Th1 CD4 T cells. Why not co-stain with those markers?
It is unclear why the authors looked at PNAd? The purpose needs to be stated explicitly. Also, a major function of HEVs is to recruit naïve T cells for activation by antigen-presenting cells. It appears an incomplete investigation to assess PNAd without looking at naïve T cells.
Author Response
The following is our response to the reviewer comments:
Reviewer: 1
In this manuscript, Elewa et al. a mouse model (Lpr) of systemic lupus erythematosus (SLE), to examine the effect of dexamethasone on histological features in the affected lung and MFALCs (mediastinal fat-associated lymphoid clusters), including lung/lymphoid tissue injury, immune infiltration, and vasculature.
My biggest concern is how this study will elucidate impactful mechanisms of SLE or novel alternative therapeutic strategies for SLE treatment. Moreover, the figure arrangement and logical flow feel like showing us a check list of scientific findings rather than a scientific story. In many figures little rationale was provided to tell readers which experiments were to address what scientific questions, and how they fit into a bigger picture.
Our response:
- Firstly, I would like to thank you for your valuable comments. We revised the manuscript based on the reviewer comments and showed the rationale and the scientific purpose.
- We added the sentence addressing what scientific questions and rationale for different experiments in the revised manuscript as following: Page3 (lines 103-106, 129-130); page 4 (lines 150-154) ; page 6 (lines 225-226); page 8 (lines 273-275); page 9 (lines 284-285).
- I also have listed below some specific concerns:
- CD68 is a pan-macrophage/monocyte marker, not specific to M1s.
- Iba1 is not a lung macrophage marker. On one hand it can be expressed on DCs; on the other hand many lung macrophages, especially CD11b-negative resident macrophages do not express Iba1.
Our response:
- We agree with the reviewer that CD68 in some publications has been reported as a pan-macrophage marker if it is used as sole marker. Simultaneously, other several reports have considered CD68 as M1 macrophage marker if co-stained with other macrophage markers (Rakaee et al., 2019).
- For Iba1, several recent reports have considered that Iba1 as pan-macrophage marker in the lung tissue (Yamauchi et al., 2019; Horkowitz et al., 2020) as well as in other organs. Furthermore, our previous reports identified Iba1 positive macrophages in both MFALCs, and lung tissue of healthy and autoimmune disease mice strains (Elewa et al., 2016, 2018).
Therefore, in the present study we used Iba1 as pan macrophage marker, and the Co-stained CD68 as M1 macrophage and the Co-stained CD204 as M2 macrophages.
Please find the refernces of previous reports:
- Yamauchi, K.; Kasuya, Y.; Kuroda,; Tanaka, K.; Tsuyusaki, J.; Ishizaki, S.; Matsunaga, H.; Iwamura, C.; Nakayama, T.; Tatsumi, K., Attenuation of lung inflammation and fibrosis in CD69-deficient mice after intratracheal bleomycin. Respir Res 2011, 12, (1), 131-131.
- Horkowitz, A. P.; Schwartz, A. V.; Alvarez, C. A.; Herrera, E. B.; Thoman, M. L.; Chatfield, D. A.; Osborn, K. G.; Feuer, R.; George, U. Z.; Phillips, J. A., Acetylcholine Regulates Pulmonary Pathology During Viral Infection and Recovery. ImmunoTargets and therapy 2020, 9, 333-350.
- Pierezan, F.; Mansell, J.; Ambrus, A.; Rodrigues Hoffmann, A., Immunohistochemical expression of ionized calcium binding adapter molecule 1 in cutaneous histiocytic proliferative, neoplastic and inflammatory disorders of dogs and cats. Journal of comparative pathology 2014, 151, (4), 347-51.
- Rakaee, M.; Busund, L.-T. R.; Jamaly, S.; Paulsen, E.-E.; Richardsen, E.; Andersen, S.; Al-Saad, S.; Bremnes, R. M.; Donnem, T.; Kilvaer, T. K., Prognostic Value of Macrophage Phenotypes in Resectable Non-Small Cell Lung Cancer Assessed by Multiplex Immunohistochemistry. Neoplasia 2019, 21, (3), 282-293.
- We added the reference indicating the use of Iba1 as lung macrophages marker and CD68 as M1 macrophage marker in the revised manuscript in pages 19-20 Lines 722-733. Also we added the citation of such references in the revised manuscript page 15 (line 515, 537).
2- The degree of proliferation and interpretation is confusing. To address what cells are proliferating, dual staining is necessary. It is unclear if Figure 3D was quantified by calculating the ratio of Ki67 positivity over immune cell percentage (which is not an appropriate analysis) or quantifying the number of Ki67-expressing leukocytes (if so, example of staining should be shown).
Our response:
- We added the dual staining for proliferating cells along with different immune cell types in supplementary figures 3-5.
- We also showed that in the revised manuscript in page 6, lines 212-221.
- We revealed that the lung of saline group showed more proliferating B lymphocytes and diminished proliferating granulocytes as compared to the dexamethasone group. However, no obvious difference in the proliferating macrophages among both studied groups.
- For detecting the proliferating T- lymphocytes, it was not possible to perform dual immunofluorescent because the IgG type for the antibodies for both Ki67 and CD3 antibodies are produced in the same species (rabbit). Simultaneously, we performed single immunostaining for the Ki67 and CD3 on serial sections and compared, but no difference could be observed in the proliferating T lymphocytes among studied groups.
3- Lyve1 is not only on lymphatic endothelium, but also on macrophages, especially M2s. Co-staining with CD31 or CD68 is needed.
Our response:
We added the co-staining for LYVE1 and CD68 in figure 5. Furthermore, we showed that in the revised manuscript in pages 7&8 (lines 248-268); page 15 (lines 541-543).
4- It is unclear what was the purpose to assess Lyve1. Recruitment of effector leukocytes to the lung relies on blood vessels, not lymphatic vessels. Blood vessels are the logical candidate of interest here.
Our response:
- According to our previous reports we revealed connection of mediastinal fat tissue with lung lobules furthermore, we reported the role of Lymphatic vessels for immune cells migration between the MFALCs and lung infiltration. Interestingly, our results revealed several lymphatic vessels engorged with immune cells in the lung of saline group than that of the Dexamethasone group (figure 5). Furthermore, we clarified Ki67 positive endothelial cells lining the lymphatic vessels in lung of saline group than that of the Dexamethasone group supplementary figure 3.
- We added data indicating that in the revised manuscript in pages 7&8 (lines 248-268).
5- Along the same line, it is unclear how co-staining of CXCL9 with Lyve1 is informative. CXCL9 attracts CD8 and Th1 CD4 T cells. Why not co-stain with those markers?
Our response:
- We added co-staining of CXCL9 along with either CD8 and CD4 T cells in the revised manuscript (supplementary figure 6).
- We showed data indicating that in the revised manuscript in pages 8&9 (lines 273-283), page 13 (lines 430-436).
It is unclear why the authors looked at PNAd? The purpose needs to be stated explicitly. Also, a major function of HEVs is to recruit naïve T cells for activation by antigen-presenting cells. It appears an incomplete investigation to assess PNAd without looking at naïve T cells.
Our response:
We agree the reviewer that the function of HEVs in general is to recruit naïve T cells and memory cells in lymphoid organs specially secondary lymphoid organs. However, the role of HEVs has been previously revealed for development of inflammation in other non-lymphoid organs (Blanchard and Girard 2021). Also, our previous reports revealed their role in the development of MFALCs and lung injury in several mice models (Elewa et al. 2018, 2021). Therefore, in the current investigation, we considered examination of their occurrence among studied groups.
- Blanchard L. and Girard, J-P. (2021): High endothelial venules (HEVs) in immunity, inflammation and cancer. Angiogenesis (2021) 24:719–753.
Reviewer 2 Report
In this manuscript, Elewa and colleagues aimed to investigate the role of dexamethasone (dex), an anti-inflammatory corticosteroid on auto-immune lung injury and mediastinal fat associated lymphoid cluster. By using MRL/lpr female mice, the authors demonstrated that lung injury and lymphoid cluster formation in the lungs as well as in the mediastinal fat was inhibited upon dexamethasone treatment. Mechanistically, dex treatment inhibit the proliferation of immune cells and formation of HEV. I have the following comments.
1-The authors measured anti-dsDNA antibodies in the serum of both saline and dex treated mice. The authors should measure additional parameter such as anti-RO/ anti-SM antibodies to confirm the autoantibody generation.
2-The frequency of immune cells were measured by morphometric analysis. FACS analysis of Lungs could reveal the absolute number of each cells.
3-In fig.3, the authors claimed that there were fewer Ki67+cells in the lung whereas there is increased granulocytes in the lung. The authors should costain Ki67 along with immune markers to ensure the particular immune cells having diminished proliferation.
4-Since the authors claimed that there are less cxcl3+ cells , the authors should score cxcr5 as well.
Author Response
Reviewer 2 comments
In this manuscript, Elewa and colleagues aimed to investigate the role of dexamethasone (dex), an anti-inflammatory corticosteroid on auto-immune lung injury and mediastinal fat associated lymphoid cluster. By using MRL/lpr female mice, the authors demonstrated that lung injury and lymphoid cluster formation in the lungs as well as in the mediastinal fat was inhibited upon dexamethasone treatment. Mechanistically, dex treatment inhibit the proliferation of immune cells and formation of HEV. I have the following comments.
- The authors measured anti-dsDNA antibodies in the serum of both saline and dex treated mice. The authors should measure additional parameter such as anti-RO/ anti-SM antibodies to confirm the autoantibody generation.
Our response:
Firstly, I would like to thank you for your valuable recommendation. We agree with the reviewer that analysis of autoantibody generation is an interestingly point. However, as reviewer know that the purpose of the current investigation is to focus on the analysis of the impact of dexamethasone on the degree of lung injury and MFALCs development, so we examined the histopathology of both lung and MFALCs among both studied groups. Furthermore, we showed the serum level of autoantibodies in the current study as final product for indication of the degree of systemic autoimmunity. Unfortunately, we could not perform such measurements because currently we did not have such kits and as the reviewer knows, that ordering some kits, and chemicals could need several months to receive due to the multiple lockdowns specially for items exported from outside Japan because of COVID situation. Again, we would like to thank the reviewer and we would like to consider analysis of such parameter in our further study.
- The frequency of immune cells were measured by morphometric analysis. FACS analysis of Lungs could reveal the absolute number of each cells.
Our response:
We agree with the reviewer that FACS analysis of Lungs could reveal the absolute number of each cell types. However, in the current investigation we would like to show the impact of dexamethasone on the histopathological changes and immune cells distribution in both lung and MFALCs. Furthermore, we performed histoplanimetrical measurement for the ratio of different immune cells. We will include the FACS analysis in our future experiment and investigation.
- In fig.3, the authors claimed that there were fewer Ki67+cells in the lung whereas there is increased granulocytes in the lung. The authors should costain Ki67 along with immune markers to ensure the particular immune cells having diminished proliferation.
Our response:
- We added the dual staining for proliferating cells along with different immune cell types in supplementary figures 3-5.
- We also showed that in the revised manuscript in page 6, lines 212-221.
- We revealed that the lung of saline group showed more proliferating B lymphocytes and diminished proliferating granulocytes as compared to the dexamethasone group. However, no obvious difference in the proliferating macrophages among both studied groups.
- For detecting the proliferating T- lymphocytes, it was not possible to perform dual immunofluorescent because the IgG type for the antibodies for both Ki67 and CD3 antibodies are produced in the same species (rabbit). Simultaneously, we performed single immunostaining for the Ki67 and CD3 on serial sections and compared, but no difference could be observed in the proliferating T lymphocytes among studied groups.
4-Since the authors claimed that there are less cxcl3+ cells, the authors should score cxcr5 as well.
Our response:
- We added the CXCR5 scores and immunofluorescent staining in figure 6A
- We showed in the revised manuscript in page 9 (lines 284-291), page 10 (lines 300-303), page 13 (lines 424-428).
Thank you for your consideration. I look forward to your favorable response.

Round 2
Reviewer 2 Report
No more comments.